# Pipelined Training with Stale Weights of Deep Convolutional Neural Networks

## Abstract

The growth in the complexity of Convolutional Neural Networks (CNNs) is increasing interest in partitioning a network across multiple accelerators during training and pipelining the backpropagation computations over the accelerators. Existing approaches avoid or limit the use of stale weights through techniques such as micro-batching or weight stashing. These techniques either underutilize of accelerators or increase memory footprint. We explore the impact of stale weights on the statistical efficiency and performance in a pipelined backpropagation scheme that maximizes accelerator utilization and keeps memory overhead modest. We use 4 CNNs (LeNet-5, AlexNet, VGG and ResNet) and show that when pipelining is limited to early layers in a network, training with stale weights converges and results in models with comparable inference accuracies to those resulting from non-pipelined training on MNIST and CIFAR-10 datasets; a drop in accuracy of 0.4%, 4%, 0.83% and 1.45% for the 4 networks, respectively. However, when pipelining is deeper in the network, inference accuracies drop significantly. We propose combining pipelined and non-pipelined training in a hybrid scheme to address this drop. We demonstrate the implementation and performance of our pipelined backpropagation in *PyTorch* on 2 GPUs using ResNet, achieving speedups of up to 1.8X over a 1-GPU baseline, with a small drop in inference accuracy.

## 1 Introduction

Modern Convolutional Neural Networks (CNNs) have grown in size and complexity to demand considerable memory and computational resources, particularly for training. This growth makes it sometimes difficult to train an entire network with a single accelerator (Huang et al., 2018; Harlap et al., 2018; Chen et al., 2012). Instead, the network is partitioned among multiple accelerators, typically by partitioning its layers among the available accelerators, as shown in Figure 1 for an example 8-layer network. The 8 layers are divided into 4 computationally-balanced partitions, $P_0...P_3$ and each partition is mapped to one of the 4 accelerators, $A_0...A_3$. Each accelerator is responsible for the computations associated with the layers mapped to it.

However, the nature of the backpropagation algorithm used to train CNNs (Rumelhart et al., 1986) is that the computations of a layer are performed only after the computations of the preceding layer in the forward pass of the algorithm and only after the computations of the succeeding layer in the backward pass. Further, the computations for one batch of input data are only performed after the computations of the preceding batch have updated the parameters (i.e., weights) of the network. These dependences underutilize the accelerators, as shown by the space-time diagram in Figure 2; only one accelerator can be active at any given point in time.

The underutilization of accelerators can be alleviated by *pipelining* the computations of the backpropagation algorithm over the accelerators (Huang et al., 2018; Harlap et al., 2018; Chen et al., 2012). That is, by overlapping the computations of different input data batches using the multiple accelerators. However, pipelining causes an accelerator to potentially use weights that are yet to be updated by an accelerator further down in the pipeline. The use of such *stale* weights can negatively affect the statistical efficiency of the network, preventing the convergence of training or producing a model with lower inference accuracy.

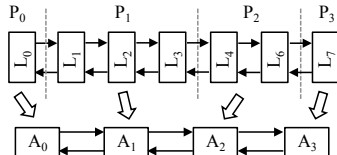
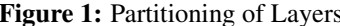

**Figure 1:** Partitioning of Layers

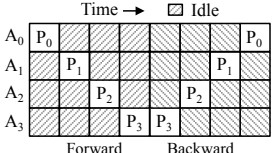

**Figure 2:** Schedule of Computations

Common wisdom is that the use of stale weights must either be avoided, e.g., with the use of micro-batches (Huang et al., 2018), be constrained to ensure the consistency of the weights within an accelerator using stashing (Harlap et al., 2018), or by limiting the use of pipelining to very small networks (Mostafa et al., 2017). However, these approaches either underutilize accelerators (Huang et al., 2018) or inflate memory usage to stash multiple copies of weights (Harlap et al., 2018).

In this paper we question this common wisdom and explore pipelining that allows for the full utilization of accelerators while using stale weights. This results in a pipelining scheme that, compared to existing schemes, is simpler to implement, fully utilizes the accelerators and has lower memory overhead. We evaluate this pipelining scheme using 4 CNNs: LeNet-5 (trained on MNIST), AlexNet, VGG and ResNet (all trained on CIFAR-10). We analyze the impact of weight staleness and show that if pipelining is limited to early layers in the network, training does converge and the quality of the resulting models is comparable to that of models obtained with non-pipelined training. For the 4 networks, the drop in accuracy is 0.4%, 4%, 0.83% and 1.45%, respectively. However, inference accuracies drop significantly when the pipelining is deeper in the network. While this is not a limitation since the bulk of computations that can benefit from pipelining are in the early convolutional layers, we address this through a hybrid scheme that combines pipelined and non-pipelined training to maintain inference accuracy while still delivering performance improvement. Evaluation shows that our pipelined training delivers a speedup of up to 1.8X on a 2-GPU system.

The remainder of this paper is organized as follows. Section 2 briefly describes the backpropagation for training of CNNs. Section 3 details our pipelining scheme. Section 4 describes how non-pipelined and pipelined backpropagation are combined. Section 5 highlights some of the implementation details. Experimental evaluation is presented in Section 6. Related work is reviewed in Section 7. Finally, Section 8 gives concluding remarks and directions for future work.

## 2 THE BACKPROPAGATION ALGORITHM

The *backpropagation* algorithm (Rumelhart et al., 1986) consists of two passes: a *forward* pass that calculates the output error and a *backward* pass that calculates the error gradients and updates the weights of the network. The two passes are performed for input data one *mini-batch* at a time.

In the forward pass, a mini-batch is fed into the network, propagating from the first to the last layer. At each layer $l$, the activations of the layer, denoted by $\mathbf{x}^{(l)}$, are computed using the weights of the layer, denoted by $\mathbf{W}^{(l)}$. When the output of the network (layer $L$) $\mathbf{x}^{(L)}$ is produced, it is used with the true data label to obtain a training error $e$ for the mini-batch.

In the backward pass, the error $e$ is propagated from the last to the first layer. The error gradients with respect to pre-activations of layer $l$, denoted by $\boldsymbol{\delta}^{(l)}$, are calculated. Further, the error gradients with respect to weights of layer $l$, $\frac{\partial e}{\partial \mathbf{W}^{(l)}}$, are computed using the activations from layer $l-1$ (i.e., $\mathbf{x}^{(l-1)}$) and $\boldsymbol{\delta}^{(l)}$. Subsequently, $\boldsymbol{\delta}^{(l)}$ is used to calculate the $\boldsymbol{\delta}^{(l-1)}$. When $\frac{\partial e}{\partial \mathbf{W}^{(l)}}$ is computed for every layer, the weights are updated using the error gradients.

In the forward pass, the activations of the layer $l$, $\mathbf{x}^{(l)}$, cannot be computed until the activations of the previous layers, i.e., $\mathbf{x}^{(l-1)}$, are computed. In backward pass, $\frac{\partial e}{\partial \mathbf{W}^{(l)}}$ can only be computed once $\mathbf{x}^{(l-1)}$ and $\boldsymbol{\delta}^{(l)}$ have been computed. Moreover, $\boldsymbol{\delta}^{(l)}$ depends on $\boldsymbol{\delta}^{(l+1)}$. Finally, for a given mini-batch the backward pass cannot be started until the forward pass is completed and the error $e$ has been determined.

The above dependences ensure that the weights of the layers are updated using the activations and error gradients calculated from the *same* batch of training data in one iteration of the backpropagation algorithm. Only when the weights are updated is the next batch of training data fed into the

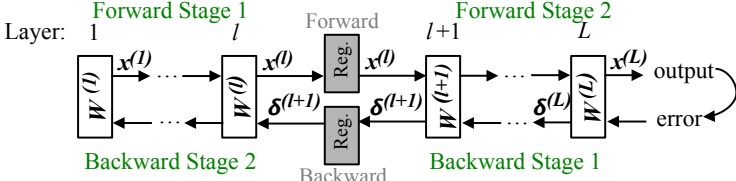

**Figure 3:** Pipelined Backpropagation Algorithm

network. These dependences limit parallelism when a network is partitioned across multiple accelerators and allow only one accelerator to be active at any point. This results in under-utilization of the accelerators. It is this limitation that pipelining addresses.

## 3   PIPELINED BACKPROPAGATION

We illustrate our pipelined backpropagation implementation with the $L$ layer network shown in Figure 3, using conceptual pipeline registers. Two registers are inserted between layers $l$ and $l + 1$; one register for the forward pass and a second for the backward pass. The forward register stores the activations of layer $l$ ($\mathbf{x}^{(l)}$). The backward register stores the gradients $\delta^{(l+1)}$ of layer $l + 1$. This defines a *4-stage* pipelined backpropagation. The forward pass for layers 1 to $l$ forms forward stage $FS_1$. The forward pass for layers $l + 1$ to $L$ form forward stage $FS_2$. Similarly, the backwards pass for layers $l + 1$ to $L$ and 1 to $l$ form backward stages $BKS_1$ and $BKS_2$ respectively.

The forward and backward stages are executed in a pipelined fashion on 3 accelerators: one for $FS_1$, one for both $FS_2$ and $BKS_1$, and one for $BKS_2$[1]. In cycle 0, mini-batch 0 is fed to $FS_1$. The computations of the forward pass are done as in the traditional non-pipelined implementation. In cycle 1, layer $l$ activations $\mathbf{x}^{(l)}$ are fed to $FS_2$ and mini-batch 1 is fed to $FS_1$. In cycle 2, the error for mini-batch 0 computed in $FS_2$ is directly fed to $BKS_1$, the activations of layer $l$ $\mathbf{x}^{(l)}$ are forwarded to $FS_2$ and mini-batch 2 is fed to $FS_1$. This pipelined execution is illustrated by the space-time diagram in Figure 4 for 5 mini-batches. The figure depicts the mini-batch processed by each accelerator cycles 0 to 6. At steady state, all the accelerators are active in each cycle of execution.

The above pipelining scheme utilizes weights in $FS_1$ that are yet to be updated by the errors calculated by $FS_2$ and $BKS_1$. At steady state, the activations of a mini-batch in $FS_1$ are calculated using weights that are 2 execution cycles old, or 2 cycles *stale*. This is reflected in Figure 4 by indicating the weights used by each forward stage and the weights updated by each backward stage. The weights of a forward stage are subscripted by how stale they are (-ve subscripts). Similarly, the weights updated by a backward stage are subscripted by how delayed they are (+ve subscripts).

Further, since the updates of the weights by $BKS_2$ requires activations calculated for the same mini-batch in $FS_1$ for all layers in the stage, it is necessary to save these activations until the error gradients with respect to the weights are calculated by $BKS_2$. Only when the weights are updated using the gradients can these activations be discarded.

In the general case, we use $K$ pairs of pipeline registers (each pair consisting of a forward register and a backward register) inserted between the layers of the network. We describe the placement of the register pairs by the *Pipeline Placement Vector*, $\text{PPV} = (p_1, p_2, ..., p_K)$, where $p_i$ represents the layer number after which a pipeline register pair is inserted. Such a placement creates $(K + 1)$ forward stages, labeled $FS_i, i = 1, 2, ..., K + 1$ and $(K + 1)$ backward stages, labeled $BKS_i, i = 1, 2, ..., K + 1$. Forward stage $FS_i$ and backward stage $BKS_{K-i+2}$ correspond to the same set of layers. Specifically, stage $FS_i$ contains layers $p_i + 1$ to $p_{i+1}$, inclusive. We assign each forward stage and each backward stage to an accelerator, with the exception of the $FS_{K+1}$ and backward stage $BKS_1$, which are assigned to the same accelerator to reduce weight staleness by an execution cycle. In total $2K + 1$ accelerators are used.

We quantify weight staleness as follows. A forward stage $FS_i$ and backward stage $BKS_{K-i+2}$ use the same weights that are $2(K - i + 1)$ cycles old. A forward stage $FS_i$ must store the activations of

---

[1]We combine $FS_1$ and $BKS_1$ on the same accelerator to reduce weight staleness.

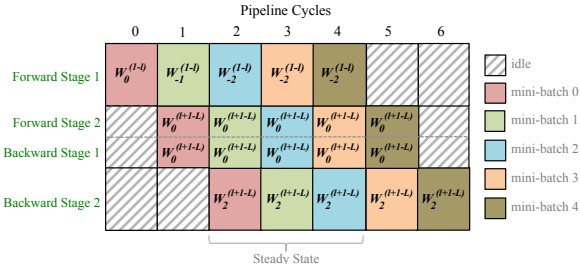

**Figure 4:** An Illustration of Computations of The Pipelined Backpropagation for Each Cycle

all layer in the stage for all $2(K-i+1)$ cycles which are used for the corresponding backward stage $\text{BKS}_{K-i+2}$. Thus, we define the *Degree of Staleness* as $2(K-i+1)$, and these saved activations as *intermediate activations*. For each pair of stages $\text{FS}_i$ and $\text{BKS}_{K-i+2}$, let there be $N_i$ weights in their corresponding layers. The layers before the last pipeline register pairs always use stale weights. Thus, we define *Percentage of Stale Weight* as $(\sum_{i=1}^{K} N_i)/(\sum_{i=1}^{K+1} N_i)$.

On the one hand, the above pipelined execution allows a potential speedup of $2K+1$ over the non-pipelined implementation, keeping all the accelerators active at steady state. On the other hand, the use of stale weights may prevent training convergence or may result in a model that has an inferior inference accuracy. Further, it requires an increase in storage for activations. Our goal is to assess the benefit of this pipelined execution and the impact of its down sides.

## 4 HYBRID PIPELINED/NON-PIPELINED BACKPROPAGATION

*Hybrid* training combines pipelined training with non-pipelined training. We start with pipelined training and after a number of iterations, we switch to non-pipelined training. This can address drops in inference accuracy of resulting models because of weight staleness, but it reduces the performance benefit since during non-pipelined training, the accelerators are under-utilized.

The extent of the speedup obtained by hybrid training with a given number of accelerators is determined by the number of iterations used for pipelined and non-pipelined training. Assume that $n_{np}$ iterations are used to reach the best inference accuracy for non-pipelined training, and that in hybrid training, $n_p$ iterations ($n_p \leq n_{np}$) are pipelined followed by $n_{np} - n_p$ iterations of non-pipelined training to reach the same inference accuracy as non-pipelined training. The speedup of hybrid training with respect to the non-pipelined training with $2K+1$ accelerators is $n_{np}/(n_p/(2K+1) + (n_{np} - n_p))$. For large $K$, then using Amdahl's law, the speedup approaches an upper bound of $n_{np}/(n_{np} - n_p)$.

## 5 IMPLEMENTATION

We implement pipelined training in two ways: *simulated* in *Caffe* (Jia et al., 2014), where the whole training process is performed on one process with no parallelism, and *actual* with parallelism across accelerators in *PyTorch* (Paszke et al., 2017). The simulated execution is used to analyze statistical convergence, inference accuracy and impact of weight staleness unconstrained by parallelism and communication overhead. The actual execution is used to report performance and *PyTorch* is used instead of *Caffe* to leverage its support for collective communication protocols and its flexibility in partitioning a network across multiple accelerators. Both *Caffe* and *PyTorch* have no support for pipelined training. Thus both were extended to provide such support.

We develop a custom *Caffe* layer in Python, which we call a Pipeline Manager Layer (PML), to facilitate the simulated pipelining. During the forward pass, a PML registers the input from a previous layer and passes the activation to the next layer. It also saves the activations for the layers connected to it to be used in the backward pass. During the backward pass, a PML passes the appropriate error gradients. It uses the corresponding activations saved during the forward pass to update weights and generate error gradients for the previous stage, using existing weight update mechanisms in *Caffe*.

| CNN | Number of Layers | 4-Stage | 6-Stage | 8-Stage | 10-Stage |
|------|------|------|------|------|------|
| LeNet-5 | 5 | (1) | (1,2) | (1,2,3) | (1,2,3,4) |
| AlexNet | 8 | (1) | (1,2) | (1,2,3) | N/A |
| VGG-16 | 16 | (2) | (2,4) | (2,4,7) | (2,4,7,10) |
| ResNet-20 | 20 | (7) | (7,13) | (7,13,19) | N/A |

**Table 1:** Pipeline Placement Vectors for CNNs

To implement actual hardware-accelerated pipelined training, we partition the network onto different accelerators (GPUs), each running its own process. Asynchronous sends and receives are used for data transfers, but all communication must go through the host CPU, since point-to-point communication between accelerators is not supported in *PyTorch*. This increases communication overhead. Similar to the PMLs in *Caffe*, the activations computed on one GPU are copied to the next GPU (via the CPU) in the forward pass and the error gradients are sent (again via the CPU) to the preceding GPU during the backward pass. The GPUs are running concurrently, achieving pipeline parallelism.

# 6 EVALUATION

## 6.1 SETUP, METHODOLOGY AND METRICS

Simulated pipelining is evaluated on a machine with one Nvidia GTX1060 GPU with 6 GB of memory and an Intel i9-7940X CPU with 64 GB of RAM. The performance of actual pipelining is evaluated using two Nvidia GTX1060 GPUs, each with 6 GB of memory, hosted in an Intel i7-9700K machine with 32 GB of RAM.

We use four CNNs in our evaluation: LeNet-5 (LeCun et al., 1998) trained on MNIST (LeCun & Cortes), AlexNet (Krizhevsky et al., 2012), VGG-16 (Simonyan & Zisserman, 2014) and ResNet (He et al., 2016), all trained on CIFAR-10 (Krizhevsky et al.). For ResNet, we experiment with different depths: 20, 56, 110, 224 and 362. We train these CNNs mostly following their original setting (LeCun et al., 1998) (Krizhevsky et al., 2012) (Simonyan & Zisserman, 2014) (He et al., 2016) with minor variations to the hyperparameters, as described in Appendix 8.

We evaluate the effectiveness of pipelined training in terms of its training convergence and its *Top-1* inference accuracy, compared to those of the non-pipelined training. We use the *speedup* to evaluate performance improvements. It is defined as the ratio of the training time of the non-pipelined implementation on single communication-free GPU to the training time of the pipelined training.

## 6.2 TRAINING CONVERGENCE AND INFERENCE ACCURACY

Figure 5 shows the improvements in the inference accuracies for both pipelined and non-pipelined training as a function of the number of training iterations (each iteration corresponds to a mini-batch). The pipelined training is done using 4, 6, 8 and 10 stages. Table 1 shows where the registers are inserted in the networks using their PPV defined in Section 3. Figure 5 shows that for all the networks, both pipelined and non-pipelined training have similar convergence patterns. They converge in more or less the same number of iterations for a given number of pipeline stages, albeit to different inference accuracies. This indicates that our approach to pipelined training with stale weights does converge, similar to non-pipelined training.

Table 2 shows the inference accuracy obtained after up to 30,000 iterations of training. For LeNet-5, the inference accuracy drop is within 0.5%. However, for the other networks, there is a small drop in inference accuracy with 4 and 6 stages. AlexNet has about 4% drop in inference accuracy, but for VGG-16 the inference accuracy drop is within 2.4%, and for ResNet-20 the accuracy drop is within 3.5%. Thus, the resulting model quality is comparable to that of a non-pipelining-trained model.

However, with deeper pipelining (i.e., 8 and 10 stages), inference accuracy significantly drops. There is a 12% and a 8.5% inference accuracy drop for VGG-16 and ResNet-20 respectively. In this case, the model quality is not comparable to that of the non-pipelined training. This results confirm what is reported in the literature (Harlap et al., 2018) and can be attributed to the use of stale weights. Below we further explore the impact of stale weights on inference accuracy.

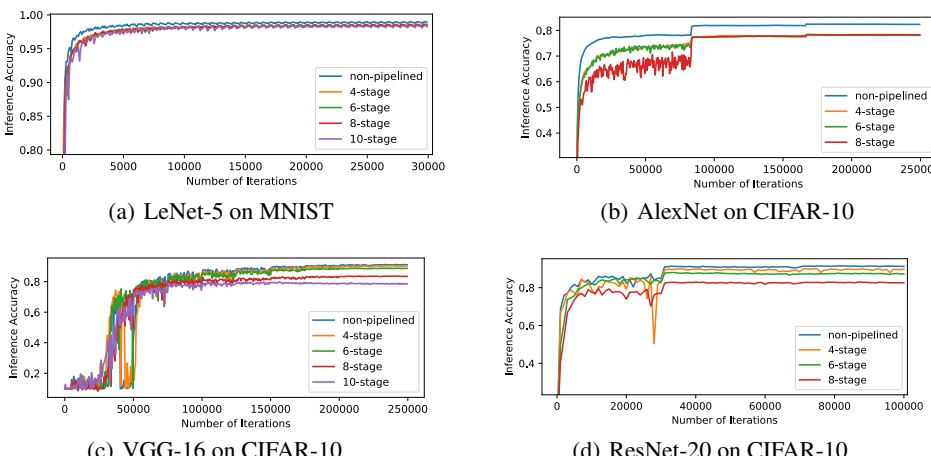

**Figure 5:** Inference Accuracy Curves for Simulated Pipelined Training

| CNN | Non-pipelined | 4-Stage | 6-Stage | 8-Stage | 10-Stage |
|---|---|---|---|---|---|
| LeNet-5 | 99.00% | 98.64% | 98.62% | 98.61% | 98.47% |
| AlexNet | 82.51% | 78.47% | 78.32% | 78.47% | N/A |
| VGG-16 | 91.36% | 90.53% | 88.96% | 83.73% | 79.85% |
| ResNet-20 | 91.50% | 90.05% | 88.00% | 83.01% | N/A |

**Table 2:** Inference Accuracy for Simulated Pipelined Training

### 6.3 IMPACT OF WEIGHT STALENESS

We wish to better understand the impact of the number of pipeline stages and the location of these stages in the network on inference accuracy. We focus on ResNet-20 because of its relatively small size and regular structure. It consists of 3 residual function groups with 3 residual function blocks within each group. In spite of this relatively small size and regular structure, it enables us to create pipelines with up to 20 stages by inserting pipeline register pairs within residual function blocks.

We conduct two experiments. In the first, we increase the number of pipeline stages (from earlier layers to latter layers) and measure the inference accuracy of the resulting model. The results are shown in Table 3, which gives the inference accuracy of pipelined training after 100,000 iterations, as the number of pipeline stages increases. The 8-stage pipelined training is created by a PPV of (3,5,7), and the subsequent pipeline schemes are created by adding pipeline registers after every 2 layers after layer 7. Clearly, the greater the number stages, the worse is the resulting model quality.

The number of stale weights used in the pipelined training increases as the number of pipeline stages increases. Thus, Figure 6 depicts the inference accuracy as a function of the percentage of weights that are stale. The curve labeled "Increasing Stages" shows that the drop in inference accuracy increases as the percentage of stale weights increases.

In the second experiment, we investigate the impact of the *degree of staleness* (Section 3). Only *one* pair of pipeline registers is inserted. The position of this register slides from the beginning of the network to its end. At every position, the percentage of stale weights remains the same as in the first experiment, but all stale weights have the same degree of staleness. The result of this experiment is shown by the curve labeled "Sliding Stage" in Figure 6. The curve shows the inference accuracy also drops as the percentage of stale weights increases. However, it also indicates that the drop of inference accuracy remains more or less the same as in the first experiment in which the degree of staleness is higher. Thus, the percentage of stale weight appears to be what determines the drop in inference accuracy and not the degree of staleness of the weights.

The percentage of stale weight is determined by where the last pair of pipeline registers are placed in the network. It is the position of this pair that determines the loss in inference accuracy. Therefore, it is desirable to place this last pair of registers as early as possible in the network so as to minimize the drop in inference accuracy.

| Stages | Inference Accuracy |
|---|---|
| Non-pipelined | 91.50% |
| 8 | 90.28% |
| 10 | 88.37% |
| 12 | 88.73% |
| 14 | 87.94% |
| 16 | 87.30% |
| 18 | 86.23% |
| 20 | 79.09% |

**Table 3:** Fine-grained Pipelining Inference Accuracy

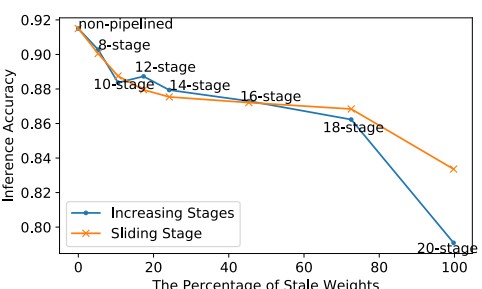

**Figure 6:** Inference Accuracy vs % of Stale Weights

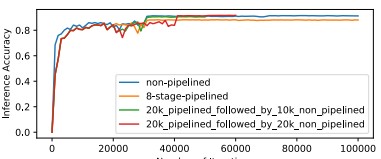

**Figure 7:** Hybrid Training Convergence

| | Inference Accuracy |
|---|---|
| Baseline 30k | 91.50% |
| Pipelined 30k | 88.29% |
| 20k+10k hybrid | 90.71% |
| 20k+20k hybrid | 91.72% |

**Table 4:** Hybrid Training Inference Accuracy

While at first glance this may seem to limit pipelining, it is important to note that the bulk of computations in a CNN is in the first few convolutional layers in the network. Inserting pipeline registers for these early layers can result in both a large number of stages that are computationally balanced. For example, our profiling of the runtime of ResNet-20 shows that the first three residual functions take more than 50% of the training runtime. This favors more pipeline stages at the beginning of the network. Such placement has the desirable effect of reducing the drop in inference accuracy while obtaining relatively computationally balanced pipeline stages.

### 6.4 EFFECTIVENESS OF HYBRID TRAINING

We demonstrate the effectiveness of hybrid training using only ResNet-20 for brevity. Figure 7 shows the inference accuracy for 20K iterations of pipelined training followed by either 10K or 20K iterations of non-pipelined training. This inference accuracy is compared to 30K iterations of either non-pipelined or pipelined training with PPV (5,12,17). The figure demonstrates that hybrid training converges in a similar manner to both pipelined and non-pipelined training. Table 4 shows the resulting inference accuracies. The table shows that the 20K+10K hybrid training produces a model with accuracy that is comparable to that of the non-pipelined model. Further, with an additional 10K iterations of non-pipelined training, the model quality is slightly better than that of the non-pipelined model. This demonstrates the effectiveness of hybrid training.

### 6.5 PIPELINED AND HYBRID TRAINING PERFORMANCE

We implement 4-stage pipelined training ResNet-20/56/110/224/362 on a 2-GPU system. Each GPU is responsible for one forward stage and one backward stage. Thus, the maximum speedup that can be obtained is 2. We train every ResNet for 200 epochs. Table 5 shows the inference accuracies with and without pipelining as model as the measured speedups of pipelined training over the non-pipelined one. The table indicates that the quality of the models produced by pipelined training is comparable to those achieved by the simulated pipelining on *Caffe*. The table further shows that speedup exists for all networks. Indeed, for ResNet-362, the speedup is 1.82X. This is equivalent to about 90% utilization for each GPU. The table also reflects that as the networks get larger, the speedup improves. This reflects that with larger networks, the ratio of computation to communication overhead is higher, leading to better speedups.

Moreover, we combine the 4-stage pipelined training described above with non-pipelined training to demonstrate the performance of hybrid training. We train every ResNet using pipelined training for 100 epochs and follow it up by 100 epochs of non-pipelined training. Because the maximum speedup for the pipelined training is 2 and only half the training epochs is accelerated, the maximum speedup for this hybrid training is $s = t/(t/2 + t/4) = 1.33$, where $t$ is the training time of non-

| ResNet | PPV | Accuracy | | | Time(Seconds) | | | Speedup | |
|---|---|---|---|---|---|---|---|---|---|
| | | Non-pipelined | Pipelined | Hybrid | Non-pipelined | Pipelined | Hybrid | Pipelined | Hybrid |
| -20 | (7) | 91.65% | 91.21% | 91.37% | 2,421 | 1,974 | 2,205 | 1.23X | 1.10X |
| -56 | (19) | 92.63% | 92.89% | 92.75% | 6,745 | 4,090 | 5,429 | 1.65X | 1.24X |
| -110 | (37) | 93.59% | 92.88% | 93.55% | 13,150 | 7,570 | 10,452 | 1.73X | 1.26X |
| -224 | (75) | 92.77% | 91.39% | 93.33% | 27,231 | 14,998 | 21,245 | 1.81X | 1.28X |
| -362 | (121) | 93.46% | 90.53% | 93.98% | 44,814 | 24,640 | 34,814 | 1.82X | 1.29X |

**Table 5:** Inference Accuracy and Speedup of Actual Pipelined and Hybrid Training

| ResNet | PPV | Activations | Weight | Increase | Increase % |
|---|---|---|---|---|---|
| -20 | (7) | 3.84MB x batch size | 1.03MB | 2.58MB x batch size | 67% |
| -56 | (19) | 10.87MB x batch size | 3.25MB | 6.32MB x batch size | 58% |
| -110 | (37) | 21.43MB x batch size | 6.59MB | 12.35MB x batch size | 57% |
| -224 | (75) | 43.70MB x batch size | 13.64MB | 25.07MB x batch size | 57% |
| -362 | (121) | 70.67MB x batch size | 22.17MB | 40.50MB x batch size | 57% |

**Table 6:** Memory Usage of 4-Stage Pipelined ResNet Training

pipelined training. Table 5 shows the inference accuracies and speedup of the hybrid training for each ResNet and validates that hybrid training can produce a model quality that is comparable to the baseline non-pipelined training while speeding up the training process. As network size grows, the speedup reaches 1.29X, approaching the theoretical limit 1.33X.

## 6.6 MEMORY USAGE

Pipelined training requires the saving of intermediate activations, as described earlier in Section 3, leading to an increase in memory footprint. This increase in memory is a function of not only the placement of the pipeline registers, but also of the network architecture and the number of inputs in a mini-batch (batch size). We calculate the memory usage of the 4-stage pipelined ResNet training above to show that this increase is modest for our pipelining scheme. Specifically, we use *torchsummary* in *PyTorch* to report memory usage for weights and activations for a network and calculate the additional memory required by the additional copies of activations. The results are shown in Table 6. Assuming a batch size of 128, the percentage increase in size is close to 60% except for ResNet-20.

## 6.7 COMPARISON TO EXISTING WORK

We compare our pipelined training scheme with two key existing systems: PipeDream (Harlap et al., 2018) and GPipe (Huang et al., 2018). We do so on three aspects: the pipelining scheme, performance and memory usage. We believe that PipeDream and GPipe are representative of existing key approaches that implement pipelined training, including Decoupled Backpropagation (DDG) (Huo et al., 2018b) and Feature Replay (FR) (Huo et al., 2018a) (discussed in Section 7).

Our pipelining scheme is simpler than that of PipeDream and GPipe in that we do not require weight stashing nor do we divide mini-batches into micro-batches. This leads to less communication overhead, and is amicable to rapid realization in machine learning framework such as *PyTorch* or in actual hardware such as Xilinx's xDNN FPGA accelerators (Xilinx, 2019).

Our pipelining scheme, as PipeDream, eliminates bubbles that exist in the pipeline leading to better performance. For example, we obtain a speedup of 1.7X for ResNet-110 using 2 GPUs in contrast to GPipe that obtains a speedup of roughly 1.3X for ResNet-101 using 2 TPUs. We also obtain similar performance compared to PipeDream for similar networks. When the number of pipeline stages grows, pipeline bubbles exhibits more negative effect on performance shown in GPipe on a 4-partition pipelined ResNet-101 using 4 TPUs as its bubble overhead doubled compared to that of the 2-partition pipelined ResNet-101.

Our scheme uses less memory compared to PipeDream, although it introduces more memory overhead compared to GPipe. PipeDream saves intermediate activations during training, as we do. However, it also saves multiple copies of a network's weights for weight stashing. The memory footprint increase due to this weight stashing depends on the network architecture, including the number of

weights and activations, as well as on the size of the mini-batch. For example, for VGG-16 trained on CIFAR-10 with a mini-batch size of 128 using a 4-stage, pipelined training, we estimate our pipelining methodology to use 49% less memory compared PipeDream. Similarly for VGG-16 trained on ImageNet (Deng et al., 2009) and a mini-batch size of 32, our scheme uses 29% less memory. We estimate the memory increase due to weight stashing also using *tourchsummary*.

## 7 RELATED WORK

There has been considerable work that explores parallelism in the training of deep neural networks. There are several approaches to exploiting parallelism.

One approach is to exploit *data parallelism* (Chen et al., 2016; Cui et al., 2016; Goyal et al., 2017; Zhang et al., 2017; Dean et al., 2012; Wang et al., 2019), in which each accelerator obtains a full copy of the model and processes different mini-batches of training data simultaneously. At the end of each training iteration, the gradients produced by all accelerators are aggregated and used to update weights for all copies of the model, synchronously (Chen et al., 2016; Goyal et al., 2017) or asynchronously (Dean et al., 2012). A centralized parameter server is usually used to facilitate data communication (Cui et al., 2016; Dean et al., 2012). Although the training is performed in parallel, the communication overhead can be significant (Wang et al., 2019).

A second approach is to exploit *model parallelism* (Kim et al., 2016; Lee et al., 2014; Chilimbi et al., 2014; Dean et al., 2012; Low et al., 2012). In this approach, a model is partitioned onto different accelerators (Kim et al., 2016; Lee et al., 2014; Low et al., 2012; Chilimbi et al., 2014; Dean et al., 2012). Each accelerator is only responsible for updating the weights for the portion of the model assigned to it. This approach is often used when a model is large and cannot fit into the memory of a single accelerator. However, because of the data dependences described in Section 2, only one accelerator is active during the training process, resulting in under-utilization of accelerators resources. Moreover, inter-layer activations and gradients across two consecutive stages needs to be communicated during training, adding more overhead to the entire process.

*Pipelined parallelism* addresses the under-utilization of accelerators resources for the training of large models. There have been a few studies that explore pipelined parallelism (Petrowski et al., 1993; Chen et al., 2012; Mostafa et al., 2017; Harlap et al., 2018; Huang et al., 2018; Huo et al., 2018b;a), which we review in this section.

PipeDream (Harlap et al., 2018) implements pipelined training for large neural networks such as VGG-16, Inception-v3 and S2VT across multiple GPUs. However, in their implementation, they limited the usage of stale weights by *weight stashing*, i.e., keeping multiple versions of network parameters (weights) during training. This increases the memory footprint of training. In contrast, we do not maintain multiple copies of weights during training, therefore reducing the memory footprint of pipelined training.

GPipe (Huang et al., 2018) implements a library in *Tensorflow* to enable pipelined parallelism for the training of large neural networks. GPipe pipelines micro-batches within each mini-batch to keep the gradients consistently accumulated. This eliminates the use of stale weight during training, but it does so at the expense of "pipeline bubbles" at steady state. GPipe utilizes these bubbles to reduce the memory footprint by re-computing forward activations instead of storing them. In contrast, our work has no pipeline bubble and thus dedicates computing resources to compute forward pass and backward pass only once during each training iteration.

Huo et al. (Huo et al., 2018b) implement decoupled backpropagation (DDG) using delayed gradient updates. They show that DDG guarantees convergence through a rigorous convergence analysis. Similar to PipeDream, DDG uses multiple copies of the weights and thus increases memory footprint. Further, DDG pipelines only the backward pass of training, leaving forward pass un-pipelined. Huo et al. (Huo et al., 2018a) follow up by proposing feature replay (FR) that re-computes activations during backward pass, similar to GPipe, resulting less memory footprint and improved inference accuracy than DDG. In contrast, we pipeline both forward and backward pass without maintaining multiple copies of weights or re-computing forward activations during backward pass.

Thus, in summary, our work contrasts to the above work on pipelined training, in that we use pipelining with unconstrained stale weights, resulting in full pipeline utilization with a modest increase in

memory usage. We extend earlier work by studying the impact of weights staleness on the quality of the model. We show that it is effective to use stale weights if the pipelining is in early layers, which is where the bulk of computations exist. Further we also extend earlier work through hybrid training, which combines both pipelined and non-pipelined training. We compare the performance and memory footprint increase of our scheme to existing work in Section 6.7.

# 8    CONCLUDING REMARKS

We evaluate pipelined execution of backpropagation for the training of CNNs in a way that fully utilizes accelerators, achieving a speedup of 1.82X on the 2-GPU system, and does not significantly increase memory usage, unlike previous work. We show that pipelining training with stale weights does converge. Further, we show that the inference accuracies of the resulting models are comparable to those of models obtained with traditional backpropagation, but only when pipelining is implemented in the early layers of the network, with inference accuracy drop within 1.45% on 4-stage pipelined training except for AlexNet. This does not limit the benefit of pipelining since the bulk of computations is in the early convolutional layers. When pipelining is implemented deeper in the network, the inference accuracies do drop significantly, but we can compensate for this drop by combining pipelined with non-pipelined training, albeit with lower performance gains, obtaining model quality with an average of 0.19% better than the baseline in inference accuracies for ResNets.

This work can be extended in a number of directions. One direction is to evaluate the approach with a larger number of accelerators since pipelined parallelism is known to scale naturally with the number of accelerators. Another is to evaluate the approach on larger datasets, such as ImageNet. Finally, pipelined parallelism lends itself to hardware implementation. Thus, another direction for future work is to evaluate pipelined parallelism using Field Programmable Gate Array (FPGA) or ASIC accelerators.

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

# A    TRAINING HYPERPARAMETERS FOR SIMULATED PIPELINED TRAINING

LeNet-5 is trained on the MNIST dataset with Stochastic Gradient Descent (SGD) using a learning rate of 0.01 with inverse learning policy, a momentum of 0.9, a weight decay of 0.0005 and a mini-batch size of 100 and for 30,000 iterations. The progression of inference accuracy during training is recorded with 300 tests

AlexNet is trained on the CIFAR-10 dataset with SGD with Nesterov momentum using a learning rate of 0.001 that is decreased by 10x twice during training, a momentum of 0.9, a weight decay of 0.004 and a mini-batch size of 100 for 250,000 iterations. One test is performed every epoch to record the progression of inference accuracy.

VGG-16 is trained on CIFAR-10 dataset with SGD with Nesterov momentum using a learning rate starting at 0.1 that is decreased by half every 50 epochs during training, a momentum of 0.9, a weight decay of 0.0005 and a mini-batch size of 100 for 250,000. Since it is relatively more difficult to train VGG-16 compared to other models, batch normalization and dropout are used during training throughout the network. One test is performed every epoch to record the progression of inference accuracy.

ResNet is trained on CIFAR-10 dataset with SGD using a learning rate starting at 0.1 and 0.01 for non-pipelined and pipelined training respectively, that is decreased by 10x twice during training, a momentum of 0.9, a weight decay of 0.0001 and a mini-batch size of 128 for 100,000 iterations. Batch normalization is used during training throughout the network. One test is performed every 100 iterations to record the progression of inference accuracy.

# B    TRAINING HYPERPARAMETERS FOR ACTUAL PIPELINED TRAINING

For the baseline non-piplined training, ResNet-20/56/110/224/362 is trained on CIFAR-10 dataset for 200 epochs with SGD using a learning rate of 0.1 that is decreased by a factor of 10 twice (at epoch 100 and 150), a momentum of 0.9, a weight decay of 0.0001 and a mini-batch size of 128. Batch normalization is used during training throughout the network. This set of hyperparameters can be found at https://github.com/akamaster/pytorch_resnet_cifar10.

For the 4-stage pipelined training, the hyperparameters are the same as the non-pipelined baseline, except for the $BKS_2$ learning rate. Table 7 shows that learning rate for all ResNet experimented.

|  | $BKS_2$ learning rate |
|---|---|
| ResNet-20 | 0.1 |
| ResNet-56 | 0.01 |
| ResNet-110 | 0.001 |
| ResNet-224 | 0.001 |
| ResNet-362 | 0.001 |

**Table 7:** Learning Rate of $BKS_2$

