# OpenReview forum: "Pipelined Training with Stale Weights of Deep Convolutional Neural Networks"
_ICLR.cc/2020/Conference — Reject_

### Official Review · AnonReviewer1 · 2019-10-23
**Official Blind Review #1**

**Rating:** 3

**Review:**

In the paper, the authors propose a pipelined backpropagation algorithm faster than the traditional backpropagation algorithm. The proposed method allows computing gradients using stale weights such that computations in different layers can be executed in parallel. They also conduct experiments to evaluate the effect of staleness and show that the proposed method is faster than compared methods. I have the following concerns:

1) There are several important works on model-parallelism and convergence guarantee of pipeline-based methods missing in this paper, for example [1][2].
2) Does the proposed method store immediate activations or recompute the activations in the backward pass?
3) In the experiments, the accuracy values are too low for me. For example, resnet110 on cifar10 is 91.99% only, it should be around 93%, an example online https://github.com/akamaster/pytorch_resnet_cifar10.
4) In the experiments, more comparisons with methods in [1] or [2] should be conducted given they are all parallelizing the backpropagation algorithm and achieve speedup in the training.
5) Last but not least, convergence analysis of the proposed method should be provided given that asynchrony may lead to divergence in the optimization.


[1] Huo, Zhouyuan, et al. "Decoupled parallel backpropagation with convergence guarantee." arXiv preprint arXiv:1804.10574 (2018).
[2] Huo, Zhouyuan, Bin Gu, and Heng Huang. "Training neural networks using features replay." Advances in Neural Information Processing Systems. 2018.


**Experience Assessment:**

I have published in this field for several years.

**Review Assessment: Checking Correctness Of Derivations And Theory:**

N/A

**Review Assessment: Checking Correctness Of Experiments:**

I carefully checked the experiments.

**Review Assessment: Thoroughness In Paper Reading:**

I read the paper at least twice and used my best judgement in assessing the paper.

---

> ### Author Response · Authors · 2019-11-11
> **Response to Reviewer's Comments**
>
> 1.	We thank the reviewer for pointing out papers [1] and [2]. We will definitely cite them in the paper and include a discussion in related work on how our scheme compares to that proposed in the two papers. In essence, our scheme is different than [1] in two key aspects: (1) we pipeline both the forward and backward passes of the backpropagation while [1] pipelines only the backward pass. Further, equation (9) in [1] suggests that while weight updates use delayed gradients, the delayed weights (W^(t-K+k)) are used for the weight gradient calculation. This is essentially similar to weight stashing used in PipeDream, which we compared to in our paper. Thus, our scheme has the advantage of a smaller memory footprint.
>
> The follow up work in [2] attempts to reduce the memory footprint through feature replay (i.e., re-computing activations during backward pass, similar to GPipe). Our scheme saves the activations instead of re-computing them to eliminate pipeline bubble, thus achieving better utilization of the accelerators (GPUs).
>
> We will edit the related work section to include the above discussion.
>
> 2.	The method proposed in our paper stores immediate activations, which is mentioned in Section 3 of the submission.
>
> 3.	We appreciate the pointer to the better performance of ResNet-110. We trained the network for only 164 epochs with a batch size of 100, which is probably the reason that its inference accuracy is lower than expected. Should we adopt the hyperparameters (a batch size of 128) and more training epochs (200 epochs) as shown at https://github.com/akamaster/pytorch_resnet_cifar10 , our ResNet-110 baseline reached 93.59% in inference accuracy, and the pipelined ResNet-110 reached 92.88% in inference accuracy. The speedup obtained is 1.73X, slightly higher than the 1.71X obtained in our paper, which could be caused by the batch size increase that makes the GPU process more efficient.
>
> The exact inference accuracy of the model is somewhat orthogonal to our study. It is the trend of the decline in inference accuracy with pipelining is what we study and this trend exists with both our hyperparameters and those at https://github.com/akamaster/pytorch_resnet_cifar10. Nonetheless, it is relatively easy for us to update the results in the paper with these new hyperparameters.
>
> 4.	Indeed, comparisons to the results in [1][2] would be interesting. However, since the scheme in [1] employ weight stashing as PipeDream does and in [2] utilizes re-computing activations, as in GPipe, our comparisons to PipeDream and GPipe subsume comparisons to [1][2], particularly given the space limitations of submission.
>
> 5.	We appreciate such detailed and rigorous convergence analysis provided in [1] and [2]. The main goal of our submission is to experimentally show that our pipelined training, using stale weights without weight stashing or micro-batching, is simpler and does converge. The paper does achieve this goal, on a number of networks. Given the limited space provided, it would be difficult to fit a convergence analysis in our paper.

---

### Official Review · AnonReviewer3 · 2019-10-26
**Official Blind Review #3**

**Rating:** 6

**Review:**

This paper investigates the impact of stale weights on the statistical efficiency and performance in a pipelined backpropagation scheme that maximizes accelerator utilization while keeping the memory overhead modest. The paper proposes to combine pipelined and non-pipelined training in a hybrid scheme to address the issue of significant drop in accuracy when pipelining is deeper in the network. The performance of the proposed pipelined backpropagation is demonstrated on 2 GPUs using ResNet with speedups of up to 1.8X over a 1-GPU baseline and a small drop in inference accuracy.

The paper is well written and easy to follow. The proposed idea is interesting and its effectiveness is well demonstrated with a promising speed and a small drop in accuracy. The proposed approach is compared to two existing works:  PipeDream [1] and GPipe [2]. Though promising results have been demonstrated, a drawback of the proposed method is that it introduces more memory overhead compared to GPipe. Although a detailed discussion is provided related to the memory consumption between the proposed method and PipeDream, no detailed discussion is provided with respect to GPipe. Further, no proper convergence analysis of the proposed approach is provided and is desired due to the likely divergence in the optimization. Minor comment: An interesting line of work is that of [3] which could be included in the discussion.

Overall, the proposed approach is interesting and is shown to achieve promising results. However, memory overhead is still an issue compared to existing method.

[1] Aaron Harlap, Deepak Narayanan, Amar Phanishayee, Vivek Seshadri, Nikhil Devanur, Greg Ganger, and Phil Gibbons. Pipedream: Fast and efficient pipeline parallel DNN training, 2018. URL http://arXiv:1806.03377.
[2] Yanping Huang, Yonglong Cheng, Dehao Chen, HyoukJoong Lee, Jiquan Ngiam, Quoc V. Le, and Zhifeng Chen. Gpipe: Efficient training of giant neural networks using pipeline parallelism, 2018. URL http://arXiv:1811.06965.
[3] Guanhua Wang, Shivaram Venkataraman, Amar Phanishayee, Jorgen Thelin, Nikhil Devanur, Ion Stoica: Blink: Fast and Generic Collectives for Distributed ML. arXiv:1910.04940, 2019.


**Experience Assessment:**

I have read many papers in this area.

**Review Assessment: Checking Correctness Of Derivations And Theory:**

I assessed the sensibility of the derivations and theory.

**Review Assessment: Checking Correctness Of Experiments:**

I carefully checked the experiments.

**Review Assessment: Thoroughness In Paper Reading:**

I read the paper thoroughly.

---

> ### Author Response · Authors · 2019-11-11
> **Response to Reviewer's Comments**
>
> Indeed, GPipe [2] incurs less memory footprint than our pipelining scheme and PipeDream [1] because it only saves the activations at the boundary of each model partition and re-computes the activations of the model during the backward pass. However, the re-computation still incurs pipeline bubbles during training. Our scheme saves all activations instead of re-computing them to eliminate pipeline bubble, thus achieving better utilization for the accelerators (GPUs). Our scheme has less memory footprint than PipeDream because it does not stash weights.
>
> The main goal of our submission is to experimentally show that our pipelined training, using stale weights without weight stashing [1] or micro-batching [2], is simpler and does converge. The paper does achieve this goal, on a number of networks. It would be difficult fit a detailed convergence analysis in our paper given the limited space provided.
>
> Thank you for pointing out paper [3]. We notice that it is submitted to arXive after the submission deadline of ICLR, thus we were unaware of it at the time of submission.  Nonetheless, we will cite it and discuss its approach in comparison to ours in the related work section of the final revised version of our paper.

---

### Official Review · AnonReviewer2 · 2019-11-02
**Official Blind Review #2**

**Rating:** 3

**Review:**

The paper proposed a new pipelined training strategy to fully utilize the memory and computational power to speed up the training process. In order to overcome the generalization degradation of the proposed method, the authors further introduced the so-called hybrid method to combine their proposed pipelined method and normal training.

The pipelined method is interesting. For the pipelined process itself, it is similar to model parallelization. For the method proposed by the paper,  it is like the async-SGD method. The paper merged these two ideas together but did not solve the problem from async-SGD, i.e. with a large number of processes, the generalization performance degrades (in the paper, it is so-called "stages"). Even with the hybrid method, the accuracy still drops.

Also, the sentence, "We demonstrate the implementation and performance of our pipelined backpropagation in PyTorch on 2 GPUs using ResNet, achieving speedups of up to 1.8X over a 1-GPU baseline, with a small drop in inference accuracy.", is confusing. If I use data parallelization, the gain should be also around 2.

The ResNet on Cifar-10 results are not convincing. The normal accuracy of ResNet20 on Cifar-10 is around 92 but the paper reported 91.1%.

Based on this, I think the paper has some room for improvement.

**Experience Assessment:**

I do not know much about this area.

**Review Assessment: Checking Correctness Of Derivations And Theory:**

N/A

**Review Assessment: Checking Correctness Of Experiments:**

I carefully checked the experiments.

**Review Assessment: Thoroughness In Paper Reading:**

I read the paper at least twice and used my best judgement in assessing the paper.

---

> ### Author Response · Authors · 2019-11-11
> **Response to Reviewer's Comments**
>
> Pipelined backpropagation is similar to model parallelism but it addresses the resource underutilization issue in model parallelism. Our pipelined method might look like async-SGD on surface. However, async-SGD (e.g. Dean et al., pointed out by Reviewer 4) utilizes data parallelism (as indicated in Dean el al.) and a parameter server to keep track of model parameters (weights). In contrast, our pipelined method does not use any parameter server. Furthermore, each accelerator obtains a replica of a full model in asycn-SGD training while each accelerator contains only a part of the model in our pipelined method, on the assumption that the full model does not fit into the memory of a single accelerator.
>
> The accuracy drops for some models in a pure pipelined training. However, hybrid training is able to bring the accuracy of most networks studied in our paper up to a comparable level of the non-pipelined baseline as shown in the evaluation section of our paper.
>
> Our pipelined method is different from data parallelism in the following way (for a 2-GPU example). For data parallelism, a model is duplicated and placed onto 2 GPUs, each GPU containing a full copy of the model. On the other hand, for pipelined parallelism, a model is divided into two partitions (on the assumption that it cannot fit in a single device): one is mapped onto GPU 0 while the other is mapped onto GPU 1, each GPU obtaining only a part of the model. Communication between these two partitions is necessary to enable activation and gradient transfers.
>
> Regardless of the parallelization techniques, the maximum speedup of a 2-GPU system is 2X compared to a 1-GPU system. To obtain a close to perfect speedup of 2X, the communication overhead must be almost non-existent and the workload needs to be perfectly balanced between the 2 GPUs. In our implementation, we obtained a speedup of 1.81X for ResNet-362, which is equivalent to 90% utilization of each GPU. Thus, our sentence the reviewer refers to.
>
> Thank you for pointing out the accuracy of ResNet-20 (similar to Reviewer 1). Again, we think the exact inference accuracy of the model is somewhat orthogonal to our study. It is the trend of the decline in inference accuracy with pipelining is what we study. This trend exists with both our hyperparameters and those at, for example, https://github.com/akamaster/pytorch_resnet_cifar10.  The use of these set of hyperparameters, obtains an inference accuracy of 91.65% (better than the accuracy stated in the original ResNet paper) for ResNet-20 non-pipelined baseline and 91.21% for pipelined version. We are not aware of any reports of an accuracy of ResNet-20 at 92% (perhaps this is approximate). Please kindly let us know a pointer. It is relatively easy to update our results in the paper with new hyperparameters.

---

### Official Review · AnonReviewer4 · 2019-11-03
**Official Blind Review #4**

**Rating:** 3

**Review:**

This paper proposes a new pipelined training approach to speedup the training for neural networks. The approach separates forward and backpropagation processes into multiple stages, cache the activation and gradients between stages, processes stages simultaneously, and then uses the stored activations to compute gradients for updating the weights. The approach leads to stale weights and gradients. The authors studied the relation between weight staleness and show that the quality degradation mainly correlates with the percentage of the weights being stale in the pipeline. The quality degradation can also be remedied by turning off the pipelining at the later training steps while overall training speed is still faster than without pipelined training.
Since this work takes the approach of allowing stale weight updates, the author should also compare with existing distributed training approaches that use asynchronous updates, with or without model parallelism, for example, Dean et al., 2012. Without the comparison it’s not clear how much improvement this approach provides compared to existing work that perform stale updates.


**Experience Assessment:**

I have read many papers in this area.

**Review Assessment: Checking Correctness Of Derivations And Theory:**

N/A

**Review Assessment: Checking Correctness Of Experiments:**

I assessed the sensibility of the experiments.

**Review Assessment: Thoroughness In Paper Reading:**

I read the paper at least twice and used my best judgement in assessing the paper.

---

> ### Author Response · Authors · 2019-11-11
> **Response to Reviewer's Comments**
>
> We thank the reviewer for pointing out the potential similarity between our pipelined approach and the asynchronous update approach. Pipelined backpropagation is similar to model parallelism but it addresses the resource underutilization issue in model parallelism. However, asynchronous update (e.g., asycn-SGD in Dean et al. [1]) usually utilizes a parameter server to keep track of model parameters (weights) while our pipelined method does not use any parameter server. Furthermore, each accelerator obtains a replica of a full model in asycn-SGD training while each accelerator contains only a part of the model in our pipelined method, on the assumption that the full model does not fit into the memory of a single accelerator.
>
> The async-SGD in Dean et al. [1] still falls into data parallelism because each accelerator has a replica of the full model. On the other hand, our approach falls into pipelined parallelism. Thus, we focused our comparison to related work on two similar approaches: PipeDream and GPipe, both utilizing pipelined parallelism. Nonetheless, we will expand the related work section to more explicitly compare to data parallelism and non-pipelined approaches to model parallelism (i.e., expand on the first paragraph of related work).
>
> [1]  Jeffrey Dean, Greg S. Corrado, Rajat Monga, Kai Chen, Matthieu Devin, Quoc V. Le, Mark Z. Mao, Marc'Aurelio Ranzato, Andrew Senior, Paul Tucker, Ke Yang, and Andrew Y. Ng. 2012. Large scale distributed deep networks. In Proceedings of the 25th International Conference on Neural Information Processing Systems

---

### Author Response · Authors · 2019-11-11
**Thanks to the reviewers**

We thank all the reviewers for their careful reading of our manuscript and their useful and constructive comments. We believe it is easy to address these comments in a slightly revised version of our manuscript and that addressing these comments increases the value and quality of our submission.

---

### Author Response · Authors · 2019-11-14
**PDF Update**

We updated the PDF of our submission to address the comments raised by the reviewers, as discussed in our responses to their comments. The updates are mostly in the related work section and minimally affect the rest of the paper describing your work. Specifically we:

1. Updated the results for the raining of the ResNets on two GPUs, based on the training hyper parameters in the link suggested by the reviewers. We now obtain similar inference accuracies as the reviews note. (Table5 on page 8).

2. Expanded the related work section to better explain the differences between data and model parallelism and to include the papers suggested by the reviewers (Section 7).

3. Justified why we limit the comparison of our experimental results to those of PipeDream and GPipe (first paragraph of Section 6.7).

4. Added Appendix B to reflect the hyper parameters used in updating the results of ResNet.

---

### Decision · Program_Chairs · 2019-12-19

**Decision:**

Reject

**Comment:**

The paper proposed a new pipelined training approach to better utilize the memory and computation power to speed up deep convolutional neural network training. The authors experimentally justified that the proposed pipeline training, using stale weights without weights stacking or micro-batching, is simpler and does converge on a few networks.

The main concern for this paper is the missing of convergence analysis of the proposed method as requested by the reviewers. The authors brought up the concern of the limited space in the paper, which can be addressed by putting convergence analysis into appendix. From a reader perspective, knowing the convergence property of the methods is much more important than knowing it works for a few networks on a particular dataset.